# Biomechanical evaluation of unilateral subcondylar fracture of the mandible on the varying materials: A finite element analysis

**Bryan Taekyung Jung**[1], **Won Hyeon Kim**[2,3], **Byungho Park**[4], **Jong-Ho Lee**[3,5],
**Bongju Kim**[3]*, **Jee-Ho Lee**[4]*

**1** School of Dentistry, University of Detroit Mercy, Detroit, Michigan, United States of America, **2** Department of Mechanical Engineering, Sejong University, Seoul, Korea, **3** Clinical Translational Research Center for Dental Science, Seoul National University Dental Hospital, Seoul, Korea, **4** Department of Oral and Maxillofacial Surgery, College of Medicine, University of Ulsan, Asan Medical Center, Seoul, Korea, **5** Department of Oral and Maxillofacial Surgery, School of Dentistry, Seoul National University, Seoul, Korea

* bjkim016@gmail.com (BK); jeehoman@gmail.com (JHL)

**Data Availability Statement:** All relevant data are within the paper.

**Funding:** This research was supported by a grant from the Korea Health Technology R&D Project

## Abstract

Fixation materials used in the surgical treatment of subcondylar fractures contribute to successful clinical outcomes. In this study, we simulated the mechanical properties of four fixation materials [titanium (Ti), magnesium alloy (Mg alloy), poly-L-lactic acid (PLLA), and hydroxyapatite/poly-L-lactide (HA-PLLA)] in a finite-element analysis model of subcondylar fracture. Two four-hole plates were fixed on the anterior and posterior surfaces of the subcondyle of the mandible. In the simulation model of a subcondylar fracture, we evaluated the stress distribution and mechanical deformation of fixation materials. The stress distribution conspicuously appeared on the condylar neck of the non-fractured side and the center of the anterior plate for all materials. More stress distribution to the biologic component appeared with HA-PLLA than with Ti or Mg alloy, but its effects were less prominent than that of PLLA. The largest deformation was observed with PLLA, followed by HA-PLLA, Mg alloy, and Ti. The results of the present study imply the clinical potential of the HA-PLLA fixation material for open reduction of subcondylar fractures.

## Introduction

Subcondylar fractures commonly occur at the sigmoid notch of the mandible. Although surgical approaches for open reduction risk facial nerve injury, they present superior results to and fewer complications than closed reduction [1] through a feasible visual field for accurate reduction with internal fixation devices such as mini-plates and screws [2]. However, the subcondyle in particular requires a very difficult surgical approach with limited surgical access due to the presence of the facial nerve. Therefore, surgeons' experience and fixation materials should be carefully considered for a successful clinical outcome. Fixation materials require mechanical strength as well as biocompatibility. Recently, with the development of biomaterials for the surgical treatment of mandible fractures, composites of unsintered hydroxyapatite

through Korea Health Industry Development Institute (KHIDI), funded by the Ministry of Health and Welfare, Republic of Korea (Grant number: HI15C1535) and a Grant (No. 2016-569) from the Asan Institute for Life Sciences, Asan Medical Center, Seoul, Korea. The funders had no role in study design, data collection and analysis, decision to publish, or preparation of the manuscript.

**Competing interests:** The authors have declared that no competing interests exist.

particles and poly-L-lactide (u-HA/PLLA) have been used for open reduction, which have led to reliable clinical outcomes in mandibular body fractures [3].

However, few studies to date have reported on HA-PLLA fixation. A previous study [4] simulated a finite-element analysis (FEA) model for open reduction internal fixation of mandibular angle fractures using four different materials to reduce the prevalence of postoperative complications: titanium (Ti), magnesium alloy (Mg alloy), biodegradable polymers such as poly-L-lactic acid (PLLA), and hydroxyapatite/poly-L-lactide (HA-PLLA). The study showed the potential clinical application of HA-PLLA for mandibular angle fractures. Extending the results from the previous study, the assessment of HA-PLLA fixation for unilateral subcondylar fracture may increase the availability of the materials in craniofacial surgery. In the present study, we used FEA to simulate a repaired unilateral subcondylar fracture and calculated the peak von Mises stress (PVMS) and deformation of mini-plates and screws of four materials: Ti, Mg alloy, PLLA, and HA-PLLA.

## Materials and methods

### Modeling of subcondylar fracture and implant system for GFEA

A mandibular bone model from a previous study was used [5]. The model consists of cortical and cancellous bones, and the thickness of the cortical bone was set to 3 mm, as described in previous literature [6, 7]. Subsequently, the subcondylar fracture was designed to run along a line obliquely connecting the sigmoid notch to the masseter tuberosity using the FEA program (ABAQUS CAE2016; Dassault Systèmes, Vélizy-Villacoublay, France). The screw and mini-plate models were constructed using a computer-aided design program (SolidWorks 2016, SolidWorks; Waltham, MA, USA). The thickness of the mini-plate was 1.0 mm, and the length and diameter of the screw were 6.0 mm and 2.0 mm, respectively (Fig 1).

The subcondylar fracture of the FEA model showed no gap between the fracture lines, assuming bone reduction (Fig 2). Two mini-plates were placed in a triangular shape on the anterior and posterior borders of the condylar neck (Fig 2). The anterior mini-plate was fixed along the coronoid notch, and the posterior mini-plate was fixed along the condylar neck (Fig 2). In our study, four types of models were considered using the material properties of Ti, Mg alloy, PLLA, and HA-PLLA, respectively. The mini-plate and screw for each type were applied with the same material properties. The bone and implant parts were assembled through the FEA program. The implant system and bone models applied were homogeneous, isotropic, and linearly elastic, according to the elastic modulus and Poisson's ratio. The mechanical properties of the bone and implant systems used were applied by referring to the literature, as presented in Table 1 [8].

Before applying the loading and boundary conditions, the cortical and cancellous bones, the mini-plate, and the screw were created using mesh generation software (Altair Hyperworks v17.0, Altair Engineering; Troy, MI, USA). The number of elements and nodes in the bone models was set to 997,961 and 207,605 for cortical bone, and 483,548 and 96,868 for cancellous bone, respectively. In the implant system, the elements and nodes were set as 172,732 and 36,089 for the screw, and 119,198 and 26,433 for the mini-plate, respectively.

### Loading and boundary conditions for masticatory motion

The rotation and movement of the two mandibular condyles were completely constrained in all directions, as reported in previous studies [9, 10]. The surfaces of the screw and bones, mini-plate and screw, and cortical and cancellous bones were applied using the tie contact condition. The tie contact condition assumed that the interfaces between the cortical and cancellous bones were fully unified or that the bone and implant systems were fully merged. The

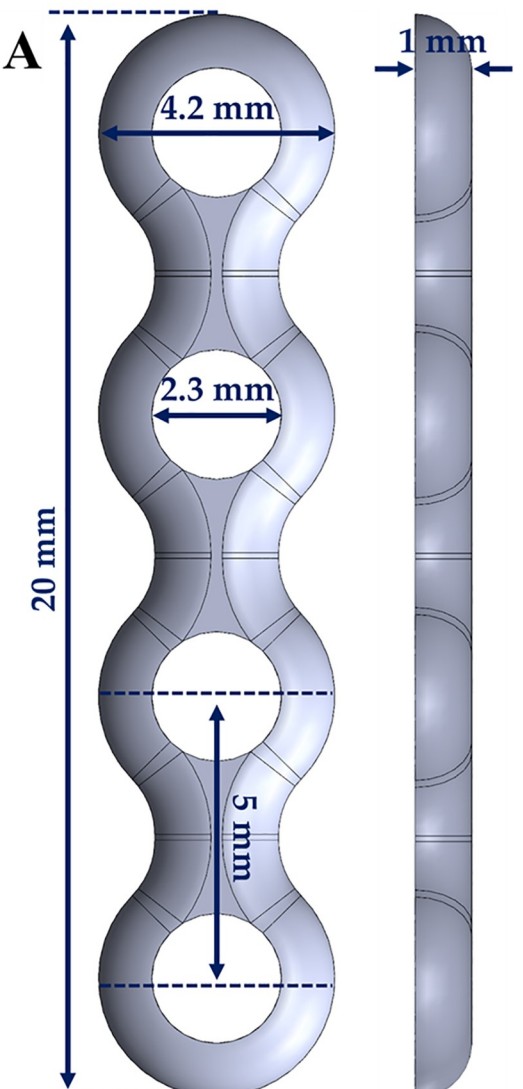

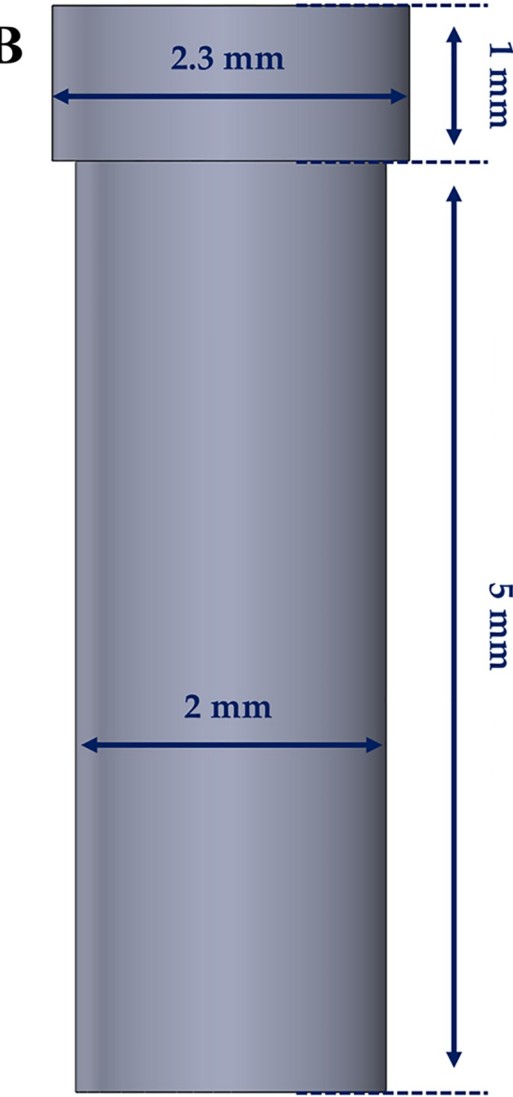

**Fig 1. Dimensions of the implant systems.** (a) mini-plate, (b) screw.

interfaces between the upper and lower sides of the fracture were assumed to be in a sliding state with a friction coefficient of 0.5 [11]. Masticatory loading was applied to the left first molar with a single node, and the loading range was set from 132–1,000 N. Starting at a load of 200 N, the load was increased at intervals of 100 N until reaching 1,000 N (Fig 2). Maximum stress distributions (MPa) and deformation (mm) were measured for the screws and mini-plates. In the cortical and cancellous bones, the maximum stress and tensile stress distributions were measured. The deformation of the implants and the distance of the fracture gap were measured for every load step ranging from 132 N to 1,000 N.

Our study used the FEA method to construct different models according to design factors and measured stress distribution and displacement for comparative analysis between various models. The FEA model for medical device analysis performs comparative analysis by changing specific designs or material properties except for the same components, such as cortical

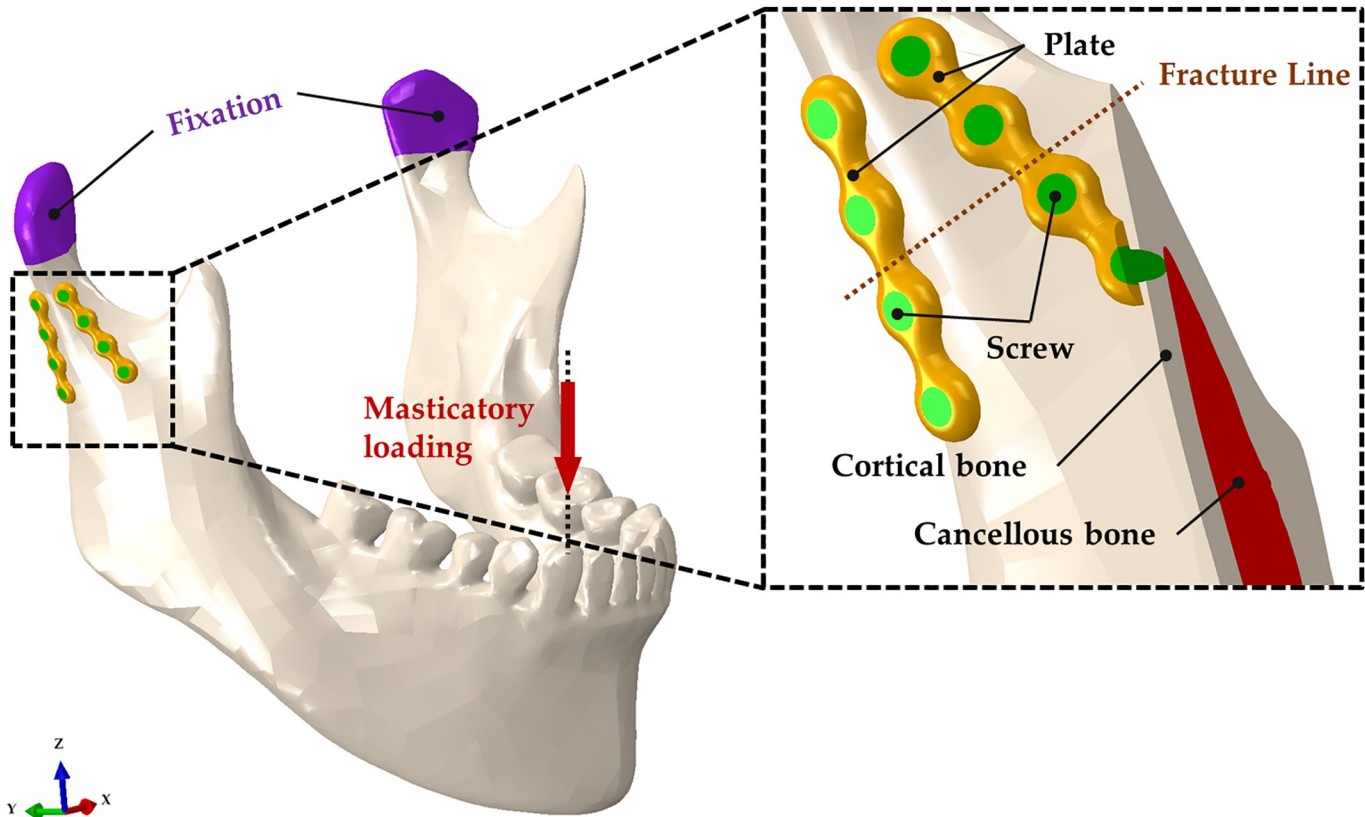

**Fig 2. Unilateral subcondylar fracture model of the mandible.** The two four-hole plates and screws are positioned at the right subcondyle while masticatory loading is applied to the non-fractured left first molar.

and cancellous bones [12–15]. Therefore, our study compared single values for each material and did not use statistical analysis.

## Results

### Tensile stress of the bones and the PVMS of the mini-plate and screw systems with four different materials

The maximal and minimal tensile stresses of the cortical and cancellous bones are shown in Tables 2 and 3 for four different materials. In the Ti system at 132 N of masticatory loading,

Table 1. Mechanical properties of implants and bones in finite-element analysis.

| Types | Elastic Modulus (MPa) | Poisson's Ratio |
|---|---|---|
| Titanium | 96,000 | 0.36 |
| Magnesium alloy | 45,000 | 0.29 |
| PLLA (biodegradable)[a] | 3,150 | 0.46 |
| HA-PLLA[b] | 9,701 | 0.317 |
| Cortical bone | 15,000 | 0.33 |
| Cancellous bone | 1,500 | 0.3 |

[a]Poly-L-lactic acid
[b]Hydroxyapatite particle/poly-L-lactide

**Table 2. Maximal and minimal tensile stresses (MPa) of cortical bone by four different materials at various masticatory loadings on subcondylar fracture.**

| Masticatory loading | Titanium | | Magnesium alloy | | PLLA | | HA-PLLA | |
|---|---|---|---|---|---|---|---|---|
| | Max[a] | Min[b] | Max[a] | Min[b] | Max[a] | Min[b] | Max[a] | Min[b] |
| 132 N | 202.1 | -24.8 | 206.3 | -31.4 | 144.5 | -27.1 | 158.1 | -34.2 |
| 200 N | 304.2 | -37.4 | 310.1 | -47.6 | 216.3 | -41.5 | 235.9 | -51.9 |
| 300 N | 451.7 | -55.9 | 459.7 | -71.5 | 319.0 | -63.1 | 346.3 | -78.1 |
| 400 N | 596.3 | -74.4 | 605.6 | -95.4 | 418.3 | -85.0 | 451.9 | -104.4 |
| 500 N | 738.0 | -92.8 | 748.1 | -119.5 | 514.3 | -107.2 | 553.2 | -130.7 |
| 600 N | 876.8 | -111.2 | 887.1 | -143.6 | 607.1 | -129.5 | 650.4 | -157.0 |
| 700 N | 1012.8 | -129.5 | 1022.8 | -167.8 | 696.9 | -151.9 | 743.7 | -183.3 |
| 800 N | 1146.1 | -147.7 | 1155.3 | -192.0 | 783.9 | -174.2 | 833.3 | -209.4 |
| 900 N | 1276.7 | -165.8 | 1284.5 | -216.2 | 868.0 | -196.3 | 919.5 | -235.3 |
| 1,000 N | 1404.7 | -183.9 | 1410.7 | -240.5 | 949.5 | -218.4 | 1002.5 | -261.0 |

[a]Maximal tensile stress

[b]Minimal tensile stress

HA-PLLA, hydroxyapatite/poly-L-lactide; PLLA, poly-L-lactic acid.

the maximal and minimal tensile stress of cortical bone were 202.1 MPa and -24.8 MPa, respectively, and those of cancellous bone were 5.76 MPa and -0.83 MPa, respectively. The maximal and minimal tensile stresses of cortical bone in the Mg alloy system were 206.3 MPa and -31.4 MPa, respectively, and those of cancellous bone were 6.69 MPa and -0.67 MPa, respectively. The maximal and minimal tensile stresses of cortical bone in the PLLA system were 144.5 MPa and -27.1 MPa, respectively, and those of cancellous bone were 4.77 MPa and -0.16 MPa, respectively. The maximal and minimal tensile stresses of cortical bone in the HA-PLLA system were 158.1 MPa and -34.2 MPa, respectively, and those of cancellous bone were 4.77 MPa and -0.16 MPa, respectively.

In each of the four different materials, the stress concentration phenomena of the cortical and cancellous bones were the same in the left mandibular condylar neck (Fig 3a, 3d, 3g and 3j). The conspicuous stress concentration appeared at the left condylar neck in the PLLA

**Table 3. Maximal and minimal tensile stresses (MPa) of cancellous bone by four different materials at various masticatory loadings on subcondylar fracture.**

| Masticatory loading | Titanium | | Magnesium alloy | | PLLA | | HA-PLLA | |
|---|---|---|---|---|---|---|---|---|
| | Max[a] | Min[b] | Max[a] | Min[b] | Max[a] | Min[b] | Max[a] | Min[b] |
| 132 N | 5.76 | -0.83 | 6.69 | -0.67 | 4.77 | -0.16 | 6.32 | -0.46 |
| 200 N | 8.68 | -1.25 | 10.06 | -1.02 | 7.16 | -0.23 | 9.45 | -0.68 |
| 300 N | 12.90 | -1.86 | 14.91 | -1.51 | 10.61 | -0.35 | 13.90 | -1.01 |
| 400 N | 17.03 | -2.46 | 19.64 | -2.01 | 13.96 | -0.46 | 18.15 | -1.31 |
| 500 N | 21.07 | -3.06 | 24.25 | -2.49 | 17.24 | -0.56 | 22.23 | -1.61 |
| 600 N | 25.02 | -3.64 | 28.73 | -2.97 | 20.43 | -0.66 | 26.12 | -1.89 |
| 700 N | 28.89 | -4.22 | 33.08 | -3.44 | 23.55 | -0.74 | 29.85 | -2.15 |
| 800 N | 32.67 | -4.79 | 37.33 | -3.90 | 26.59 | -0.81 | 33.43 | -2.41 |
| 900 N | 36.37 | -5.35 | 41.46 | -4.36 | 29.56 | -0.86 | 36.84 | -2.64 |
| 1,000 N | 39.99 | -5.91 | 45.49 | -4.81 | 32.45 | -0.92 | 40.10 | -2.87 |

[a]Maximal tensile stress

[b]Minimal tensile stress

HA-PLLA, hydroxyapatite/poly-L-lactide; PLLA, poly-L-lactic acid.

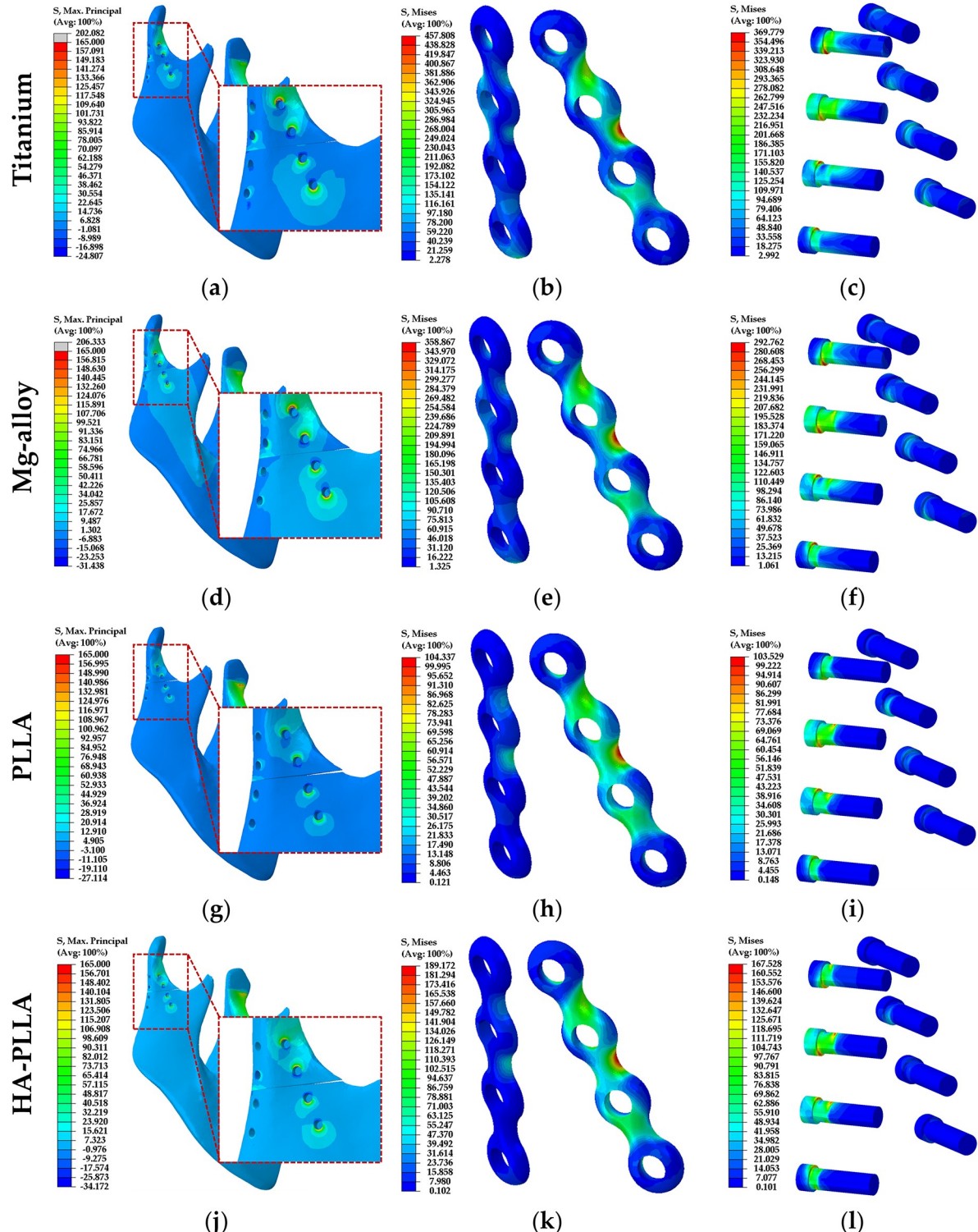

**Fig 3. Distribution of stress in the subcondylar fracture model according to material.** Left column, stress on mandibular bones; middle column, stress on the mini-plates; right column, stress on the screws. HA-PLLA, hydroxyapatite/poly-L-lactide; PLLA, poly-L-lactic acid.

**Table 4. Peak von Mises stresses of different fixation systems with 132 N of masticatory loading on subcondylar fracture.**

| Types | Titanium | Magnesium alloy | PLLA | HA-PLLA |
|---|---|---|---|---|
| Plate | 457.81 | 358.87 | 104.34 | 189.17 |
| Screw | 369.78 | 292.76 | 103.53 | 167.53 |

HA-PLLA, hydroxyapatite/poly-L-lactide; PLLA, poly-L-lactic acid.

compared to other materials (Fig 3g). The tensile stress distribution of the cortical bone was highest for the Mg alloy, followed by Ti, HA-PLLA, and PLLA. The stress distribution was similar between Ti and Mg alloy and between HA-PLLA and PLLA (Tables 2 and 3). On the other hand, the tensile stress distribution of cancellous bone was highest for the Mg alloy, followed by HA-PLLA, Ti, and PLLA. The stress distribution of the screw hole in the bones was higher in the anterior screw hole than in the posterior hole. Facing the fracture line, the contact area of the screw hole experienced compressive stress, and tensile stress occurred on the opposite side (Fig 3a, 3d, 3g and 3j).

In the Ti system, the PVMS values of the mini-plate and screw were 457.81 MPa and 369.78 MPa, respectively (Table 4). In the Mg alloy system, the PVMS values of the mini-plate and screw were 358.87 MPa and 292.76 MPa, respectively (Table 4). The stress concentration was shown at the upper second screw hole of the anterior plate (Fig 3b and 3e) and the necks of all screws, which was especially conspicuous at the upper second screw (Fig 3c and 3f). In the PLLA system, the PVMS values of the mini-plate and screw were 104.34 MPa and 103.53 MPa, respectively (Table 4). The stress distribution on the implant system was similar to that on the Ti and Mg alloy systems (Fig 3h and 3i). The pattern of stress distribution on the screw was similar to that in the Ti and Mg alloy systems, but with lower intensity (Fig 3i). The PVMS values of the mini-plate and screw in the HA-PLLA system were 189.17 MPa and 167.53 MPa, respectively (Table 4). The results were remarkably lower than those for either the Ti or Mg alloy systems but higher than those for PLLA (Fig 3l). The stress distribution at the plates was similar to that in the other groups (Fig 3k), and the pattern of PVMS was similar to that of PLLA at the screw (Fig 3l).

## Deformation of materials during masticatory loading

The deformation of the mini-plates and screws with the PLLA material was higher than that of Ti, Mg alloy, or HA-PLLA. The largest deformations of the implant systems were in the order of PLLA, HA-PLLA, Mg alloy, and Ti. The deformation in all materials did not exceed 0.4 mm (Table 5 and Fig 4).

Ti and Mg alloy fixations displayed a deformation trend that was relatively linear as masticatory loading increased. In addition, the deformation of PLLA and HA-PLLA both showed a linearly increasing trend. The slope increased at 200 N in all materials (Fig 4).

## Distance of the fracture gap during masticatory loading

The largest gap distance of the fracture site was observed for PLLA, followed by HA-PLLA, Mg alloy, and Ti (Table 6 and Fig 5). The distance of the fracture gap between bones increased linearly with respect to increased masticatory loading regardless of material.

The gap distance became wider as the masticatory loading increased regardless of material (Fig 6). In all materials, contact between the posterior fracture surfaces occurred as masticatory loading increased from 132 N (Fig 5).

**Table 5. Deformation (mm) of implant systems (mini-plate and screw) under loading conditions according to the four different materials.**

| Masticatory loading (N) | Titanium | | Magnesium alloy | | PLLA | | HA-PLLA | |
|---|---|---|---|---|---|---|---|---|
| | Screw | Plate | Screw | Plate | Screw | Plate | Screw | Plate |
| 132 | 0.017 | 0.017 | 0.018 | 0.022 | 0.050 | 0.061 | 0.034 | 0.036 |
| 200 | 0.023 | 0.026 | 0.027 | 0.033 | 0.075 | 0.091 | 0.051 | 0.054 |
| 300 | 0.033 | 0.038 | 0.041 | 0.049 | 0.112 | 0.132 | 0.076 | 0.080 |
| 400 | 0.042 | 0.051 | 0.054 | 0.065 | 0.149 | 0.172* | 0.100 | 0.106 |
| 500 | 0.051 | 0.063 | 0.067 | 0.080 | 0.187* | 0.209* | 0.124 | 0.131 |
| 600 | 0.060 | 0.075 | 0.079 | 0.095 | 0.224* | 0.246* | 0.147 | 0.157* |
| 700 | 0.068 | 0.087 | 0.092 | 0.110 | 0.261* | 0.282* | 0.170* | 0.183* |
| 800 | 0.076 | 0.098 | 0.104 | 0.124 | 0.299* | 0.319* | 0.193* | 0.208* |
| 900 | 0.083 | 0.110 | 0.116 | 0.138 | 0.336* | 0.355* | 0.216* | 0.232* |
| 1,000 | 0.090 | 0.121 | 0.128 | 0.152* | 0.374* | 0.391* | 0.235* | 0.257* |

*Fixation deformation > 0.15 mm. HA-PLLA, hydroxyapatite/poly-L-lactide; PLLA, poly-L-lactic acid.

## Discussion

Ti mini-plates and screws are the standard fixation material for facial bone trauma, with many supportive results from previous studies [16]. Ti is a desirable material because of its mechanical strength and higher biocompatibility compared to other metallic materials, such as vitallium [17, 18]. Despite these advantages, Ti fixation occasionally fails due to postoperative inflammation and thermal hypersensitivity [19, 20]. Therefore, the demand from surgeons for

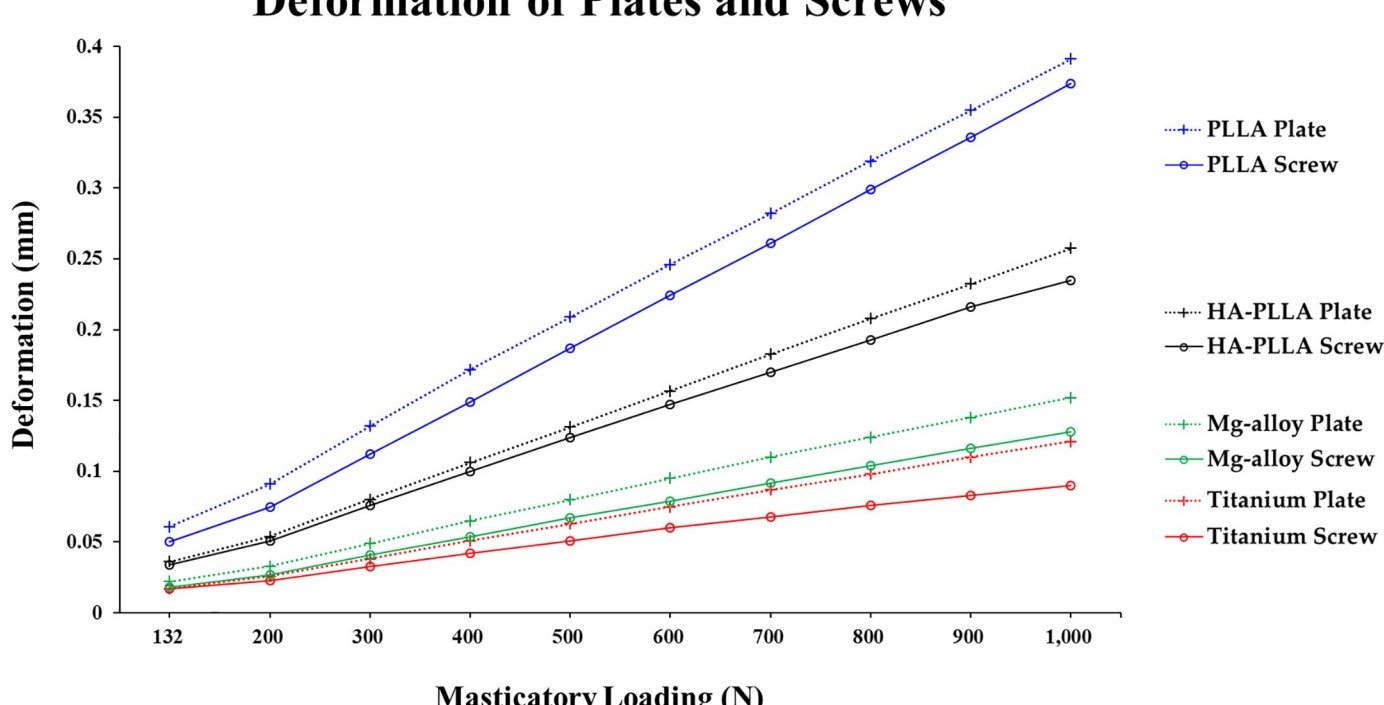

**Fig 4. Deformation of fixation materials upon masticatory loading in the subcondylar fracture simulation.** The fixation systems by all four materials showed linearly increasing deformation trends from 200 N to 1,000 N. HA-PLLA, hydroxyapatite/poly-L-lactide; PLLA, poly-L-lactic acid.

**Table 6. Gap distance (mm) between fracture surfaces under loading conditions according to four different materials.**

| Masticatory loading (N) | Titanium | Magnesium alloy | PLLA | HA-PLLA |
|:---:|:---:|:---:|:---:|:---:|
| 132 | 0.054 | 0.074 | 0.200 | 0.145 |
| 200 | 0.082 | 0.111 | 0.299 | 0.218 |
| 300 | 0.121 | 0.165 | 0.443 | 0.322 |
| 400 | 0.160 | 0.217 | 0.582 | 0.423 |
| 500 | 0.198 | 0.268 | 0.717 | 0.522 |
| 600 | 0.235 | 0.318 | 0.848 | 0.618 |
| 700 | 0.271 | 0.367 | 0.976 | 0.712 |
| 800 | 0.306 | 0.415 | 1.100 | 0.803 |
| 900 | 0.341 | 0.466 | 1.221 | 0.892 |
| 1,000 | 0.375 | 0.507 | 1.339 | 0.979 |

HA-PLLA, hydroxyapatite/poly-L-lactide; PLLA, poly-L-lactic acid.

biodegradable materials has increased because they are expected to achieve adequate stability for bony union while still resorbing in a timely fashion [16]. A previous study assessed the stability of unilateral mandibular angle fracture reduction using four different materials [4], which showed that HA-PLLA fixation could be a reasonable alternative. Extending this idea to the subcondylar region, we simulated stress and deformation of a unilateral subcondylar fracture with fixation by various materials using FEA. For the positioning of the screws and the

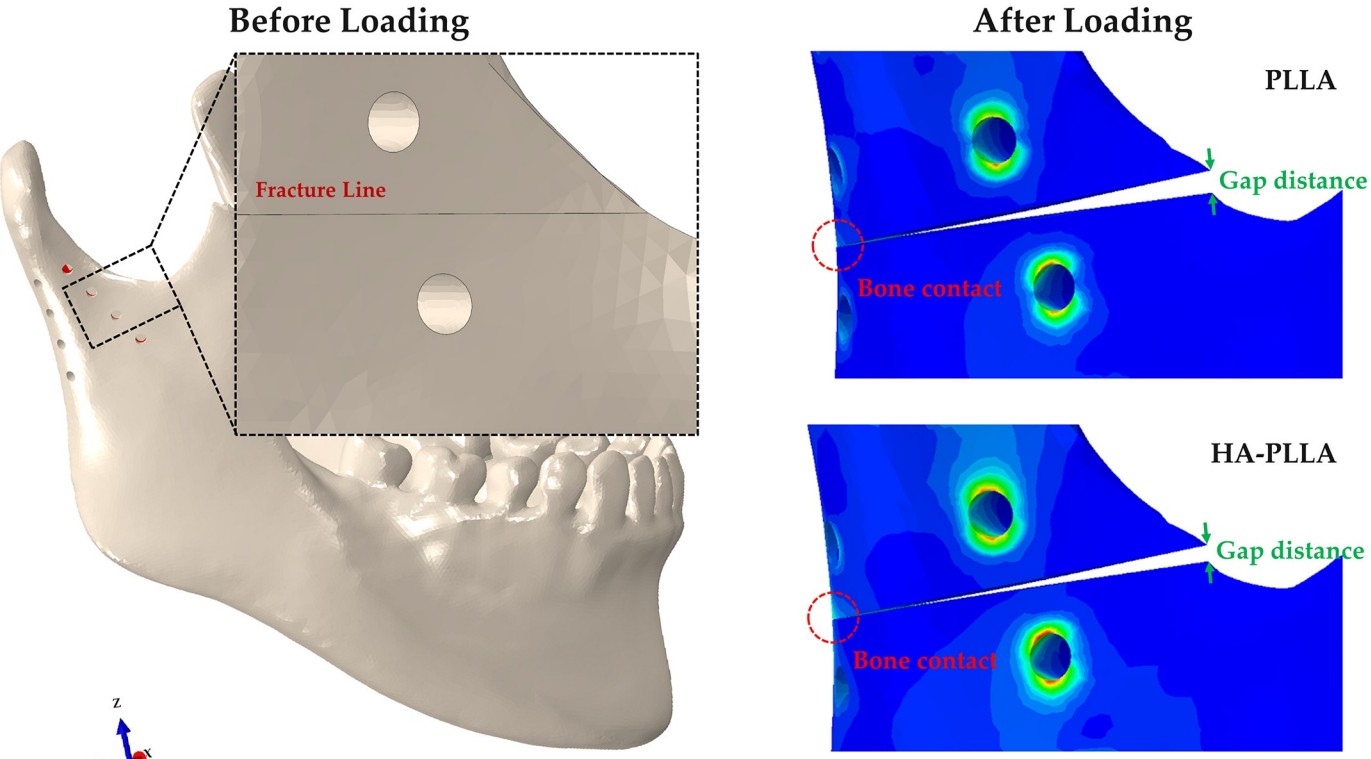

**Fig 5. Gap distance of two fragment parts upon masticatory loading in the subcondylar fracture simulation.** Regardless of material, the gap distance showed an increasing trend. HA-PLLA, hydroxyapatite/poly-L-lactide; PLLA, poly-L-lactic acid.

## Gap Distance Between Superior and Inferior Fragments

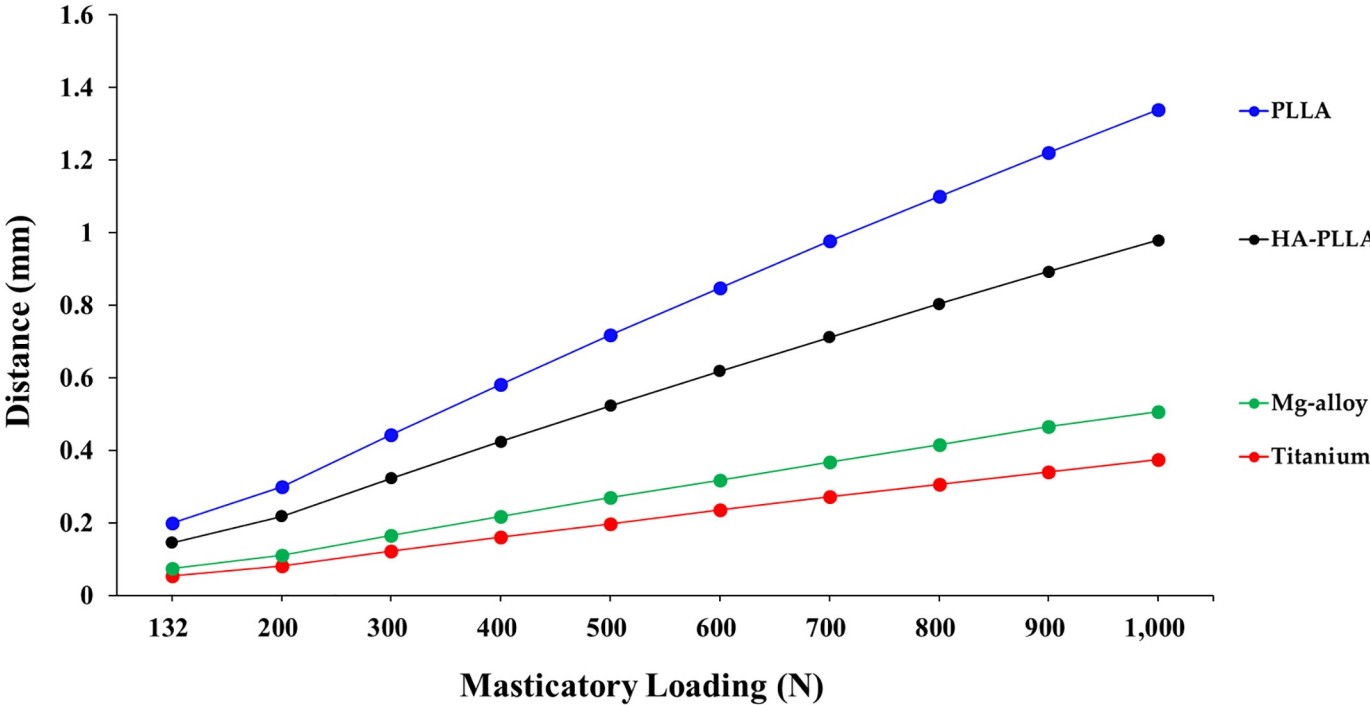

**Fig 6. Gap distance of fixation materials upon masticatory loading in the subcondylar fracture simulation.** HA-PLLA, hydroxyapatite/poly-L-lactide; PLLA, poly-L-lactic acid.

plate, previous studies demonstrated better stability using the double mini-plate fixation technique than a single-plate fixation technique for mandibular condylar fractures [21–23]. In the subcondylar area, the anterior part of the condylar process experiences tensile strains, while the posterior border experiences compressive strains [24, 25]. Ideally, two mini-plates should be placed in a triangular manner where one is applied at the posterior border of the condylar neck and the second is applied along the anterior border [24, 25]. At least two screws should be engaged on each fracture side to stabilize the fixation until it heals completely against rotation [26]. Therefore, in the FEA model of this study, the anterior four-hole mini-plate was fixed superiorly and inferiorly along the coronoid notch, while the posterior four-hole mini-plate was placed along the condylar neck and ramus. We kept anatomical conditions consistent with those of a previous study. For the mini-plate and screw design, we maintained consistency with unified concepts regardless of material in accordance with previous studies [4, 6, 7].

We assumed a force of 132 N at 1 week after surgery, 300 N at 6 weeks after surgery, and 700 N in the mastication loading of healthy adults based on previous studies [4, 10, 27]. According to these analyses, we noticed that the non-fractured side experienced more masticatory force than the fractured side during the healing period. Similar to the previous unilateral angular fracture study, we increased the masticatory loading from 132 N to 1000 N in increments of 100 N at the first molar on the non-fractured side. The upper value of the simulation in this study far exceeds the masticatory loading force of 700 N reported by Ferrario et al. [28] because our goal was to measure stress distributions and deformations of various fixation materials under extreme conditions.

In the FEA model of the mandible, stress was distributed on the condylar neck of the non-fractured side and was more conspicuous in the biodegradable PLLA and HA-PLLA models rather than the metal Ti and Mg alloy groups (Fig 3a, 3d, 3g, and 3j).

The PVMS values of the mini-plates and screw with the Ti and Mg alloy materials were over 1.5 times that of the maximal tensile stress of the cortical bone. The PVMS values of the mini-plate and screw with HA-PLLA were slightly greater than the maximal tensile stress of the cortical bone, whereas they were smaller than the maximal tensile stress of the cortical bone with PLLA (Tables 2–4). This means that metal materials, such as Ti and Mg alloy, maintained stress at the fixation components rather than transferring it to the biologic components. HA-PLLA showed relatively less stress distribution on the non-fractured side of the condyle compared to PLLA, although the biodegradable materials tended to inflict more stress than the metal materials on the biological components. Major stress was concentrated at the upper second screw hole and the center of the anterior plate, whereas less stress distribution appeared at the posterior plate regardless of the material used (Fig 3b, 3e, 3h, and 3k). The neck of the upper second screw of the anterior plate showed prominent stress concentration in all materials (Fig 3c, 3f, 3i, and 3l). These phenomena are consistent with the findings of Throckmorton and Meyer in that tensile strength was present on the anterior surface of the condyle with compressive strength on the posterior surface of the condyle [25, 26]. According to these results, modification of the anterior plate and screw design should be considered to improve clinical outcomes, which would reinforce the central part of the plate and necks of the screws.

Plates tend to experience more deformation than screws under controlled conditions. The largest deformation appeared with PLLA, followed by HA-PLLA, Mg alloy, and Ti (Fig 4). A material with a high stress value was apt to experience lower deformation (Tables 4 and 5). In our study, masticatory loading on the left side of the mandible was assumed to be from 132 N to 1000 N after surgery with increases of 100-N increments. Søballe suggested that the bone gap should be maintained within 0.15 mm to ensure good clinical outcomes during the healing period [28]. Ti presented the highest PVMS resistance and consequently had the least deformation among the four materials. Even at 1000 N (an extreme force), the displacement did not exceed 0.15 mm. The Mg alloy maintained deformation within 0.15 mm until 900 N of masticatory loading was applied. HA-PLLA exceeded 0.15 mm of deformation at 600 N. The maximum deformation of HA-PLLA was 0.157 mm at 600 N. However, PLLA showed less than 0.15 mm of deformation under 300 N (Table 5). The masticatory force of healthy adults was assumed to be 700 N [27] in the FEA model; thus, Ti and the Mg alloy could withstand normal masticatory function, although theoretically, even immediately after surgery. PLLA and HA-PLLA could maintain approximately 0.15 mm of physiologic discrepancy to 300 N, which is the masticatory force at 6 weeks after surgery [29].

According to the results of the FEA simulation, HA-PLLA would stabilize the physiologically open reduction of the subcondylar fracture until the formation of a primary callus is complete, whereas PLLA could not under 132 N (masticatory loading at 1 week after surgery) [6]. Although deformations in masticatory loading differed among the materials, no deformations exceeded 1.0 mm of discrepancy, which can be compensated by physical therapy using intermaxillary fixation [2]. However, by continuing to maintain a 0.15-mm physiologic gap during primary callus formation, HA-PLLA is clinically superior to PLLA as a biodegradable fixation material. When loading was applied to the left mandibular first molar, the gap between the superior and inferior fragments of the posterior part narrowed due to compressive stress. In this case, the compressive contact between the posterior surfaces acted like a splint. On the other hand, the gap on the anterior part of the fragments widened, creating a wedge-shaped opening due to tensile force (Fig 5). Therefore, a modified anterior plate may be needed as reinforcement to cope with tensile stress.

Compared to a previous study of mandibular angle fracture (0.136 mm in screws, 0.148 mm in plate) [4], deformation at 300 N of HA-PLLA did not show a conspicuous difference in the subcondylar fracture (0.076 mm in screw vs. 0.080 mm in plate). The results suggested that triangular fixation with two plates in the subcondylar fracture might achieve higher stability in mandibular angle fractures, although open reduction of the subcondyle is more mechanically unfavorable than that of the mandibular angle [2].

More stress distribution to biologic components occurred with HA-PLLA than with Ti or the Mg alloy, but this effect was lower than that of PLLA. HA-PLLA also showed less deformation than conventional PLLA for open reduction in the subcondylar fracture. Although the mechanical properties of HA-PLLA were inferior to those of metal materials, modification of the plate design reinforcing the central part of the anterior plate and a two-plate triangular fixation method might overcome its mechanical defects.

Subcondylar fracture is commonly unfavorable compared to mandibular body and angle fractures. This is because its dimensions are relatively small, upon which most of the mastication stress would be concentrated. Therefore, materials for subcondylar fracture should have appropriate mechanical strength as well as biocompatibility.

The results of the present study imply the clinical potential of HA-PLLA as a fixation material for open reduction of subcondylar fractures. However, this study is based on an FEA simulation model, which lacks actual biologic components. Therefore, further studies should investigate biomechanical tests and long-term clinical evaluations prior to widespread clinical application.

## Conclusion

A material with a high stress distribution tended to experience less deformation, which reflected the results of stress distribution. The deformation in HA-PLLA at 600 N was less than 0.16 mm, implying that triangular fixation with two plates in a subcondylar fracture could be expected to have a favorable clinical outcome [2, 4, 6, 28–30].

## Author Contributions

**Conceptualization:** Bryan Taekyung Jung, Byungho Park, Jee-Ho Lee.

**Data curation:** Bryan Taekyung Jung, Won Hyeon Kim, Jee-Ho Lee.

**Formal analysis:** Won Hyeon Kim, Bongju Kim.

**Investigation:** Bryan Taekyung Jung, Won Hyeon Kim, Byungho Park, Jong-Ho Lee, Bongju Kim.

**Methodology:** Won Hyeon Kim, Jong-Ho Lee, Bongju Kim.

**Project administration:** Bongju Kim, Jee-Ho Lee.

**Software:** Won Hyeon Kim.

**Visualization:** Won Hyeon Kim.

**Writing – original draft:** Bryan Taekyung Jung, Byungho Park, Jee-Ho Lee.

**Writing – review & editing:** Bryan Taekyung Jung, Won Hyeon Kim, Jong-Ho Lee, Bongju Kim, Jee-Ho Lee.

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
