## [Decision Letter · Decision Letter 0]

23 Jun 2020

PONE-D-20-14876

Biomechanical evaluation of unilateral subcondylar fracture of the mandible on the varying materials: a finite element analysis

PLOS ONE

Dear Dr. Kim,

Thank you for submitting your manuscript to PLOS ONE. After careful consideration, we feel that it has merit but does not fully meet PLOS ONE’s publication criteria as it currently stands. Therefore, we invite you to submit a revised version of the manuscript that addresses the points raised during the review process.

Please follow the reviewers comments strictly.

We look forward to receiving your revised manuscript.

Kind regards,

Essam Al-Moraissi

Academic Editor

PLOS ONE

Journal Requirements:

Reviewers' comments:

Reviewer's Responses to Questions

**Comments to the Author**

1. Is the manuscript technically sound, and do the data support the conclusions?

Reviewer #1: Yes

Reviewer #2: Yes

2. Has the statistical analysis been performed appropriately and rigorously? 

Reviewer #1: No

Reviewer #2: N/A

3. Have the authors made all data underlying the findings in their manuscript fully available?

Reviewer #1: No

Reviewer #2: Yes

4. Is the manuscript presented in an intelligible fashion and written in standard English?

Reviewer #1: Yes

Reviewer #2: Yes

5. Review Comments to the Author

Reviewer #1: Thank you for giving me this opportunity to review this in vitro research article entitled, "Biomechanical evaluation of unilateral subcondylar fracture of the mandible on the

varying materials: a finite element analysis".

I here carefully reviewed the submitted set of the manuscript and found it possibly merits of publication. However, the article needs substantial revisions in order to meet the standard of scientific publication. Therefore, I would like to propose that the article should be revised thoroughly and need further reconsideration for re-evaluation whether this study article should be appropriate for acceptance or not.

1. What is the backgrond of this in vitro study. The ORIF of mandibular condyle fractures are well documented with some proved clinical evidence so far with various surgical approaches and various hardware materials. The back ground of this study must be well discribed using appropriate and recent references. The Introduction section and the Discussion sections are to be both the level of clinical reports of undergraduates, I'm sorry but, I need to say. The authors here also need to mention the ideal lines of double buttressing of mandibular condyle fracture should be well addressed in this section.

2. In the M&M section, all the materials tested for this research should be summarized including plate thickness, screw diameter, etc. "The thickness of the mini-plate was 2.0 mm, and the length and radius of the screw were 6.0 mm and 1.0 mm, respectively." is very unclear. The authors need to carry these photos as a figure. 1.0mm screw radius means 2.0mm diameter?? The surgeons don't describe the screw system with radius!

3. "The subcondylar fracture of FEA model was constructed that a 0.5 mm gap was set to minimize the effect of shear stress caused by the bony contact" seems hardly understandable. The authors need to further discuss this issue in the Discussion section well to persuade this idea. Actually the fractured segments are honestly well reduced as "buttressing as anatomical reduction" and the fixation should be done in a clinical setting. The present idea in in vito study should be against the clinical setting completely.

4. Statistical analyses are mandatory. I've never reviewed the article without any statistical evaluation like this article, I'm afraid.

5. The discussion sections should be by far well discussed mentioning and comparing the results obtained here with those reported previously and with the clinical issues in a clinical setting. The comprehensive structures of the Discussion should be further amended. The clinical relevance should be well discribed and discussed based on the results obtained here.

Could you please let me further re-review after substantial revisions are to be done for reconsideration for suitability?

Reviewer #2: 1. Summary of the research and my overall impression:

The aim of this research is particularly relevant to the field of Craniomaxillofacial Surgery because it is related to oral fractures, more specifically to mandible condylar ones. This subject is really challenging. Sometimes, surgeons don’t choose the open access to treat these fractures either due to anatomic reasons because important structures can be found there or due to the difficulty to reduce and to fix the fractures.

It is known that the open treatment is better than the closed one because it can facilitate the return to normal activities such as mastication and occlusion as soon as possible. Besides that, some complications caused by the non-surgical treatment can be avoided.

This paper shows a study about four kinds of ORIF to treat the mandible subcondylar fractures: titanium (Ti), magnesium alloy (Mg alloy), biodegradable polymers such as poly-L-lactic acid (PLLA) and a hydroxyapatite/poly-L-lactide (HA-PLLA). It is justified by the increasing demand of re-absorbable materials to be used instead of titanium, which is the gold standard material, owing to some disadvantages.

The methodology is especially interesting, based on other studies that have been published in high impact factor journals.

2. Discussion of specific areas for improvement

Abstract:

In order to be better understood by the readers of the journal, it would be interesting to follow the acronyms with the complete name of the materials. For example ti: titanium.

I suggest removing this sentence: “Deformation of HA-PLLA was approximately 0.15 mm upon 300 N of masticatory loading”. It is hard to understand in this section and it is well explained in the results section.

Introduction:

Line 10: “Subcondylar fractures can be treated conservatively or surgically”.

What does “conservatively” mean? The best term may be “non-surgically”, as we can have a conservative surgery, for example, depending on our care during the procedure.

Materials and methods:

In Figure 1, the posterior plate does not seem to be completely attached to the bone. I would like you to confirm this and try to explain it better.

The second screws (considering the superior to inferior) are closer to the fracture line than the ideal. On the anterior plate, this screw seems to be in the fracture line. Depending on the direction of the perforation to install this screw, I think that the fracture can be “open”. Maybe, this fact can be considered a bias.

I don’t think it can influence the results, but I think that you can try to reinstall this plate, virtually.

I suggest you provide more information about the masticatory load that was applied. Why did you choose this load? Isn´t it too high? I would like to see other papers that justify this choice.

Results:

In Figure 2: “middle column, stress on fixations (mini-plates and screws) but we can just see the plates, without screws”.

I think that you need to change this caption due to the fact that the screws are not there, just the plates.

Do you have information about the bone? If so, you can insert main maximum tension and minimum tension to increase the amount of information and to make this article even more relevant.

6. PLOS authors have the option to publish the peer review history of their article (what does this mean?). If published, this will include your full peer review and any attached files.

Reviewer #1: No

Reviewer #2: Yes: Ricardo Augusto Conci

---

## [Author Response · Author response to Decision Letter 0]

12 Aug 2020

#Reviewer1

Comments and suggestions

Thank you for giving me this opportunity to review this in vitro research article entitled, "Biomechanical evaluation of unilateral subcondylar fracture of the mandible on the varying materials: a finite element analysis".

I here carefully reviewed the submitted set of the manuscript and found it possibly merits of publication. However, the article needs substantial revisions in order to meet the standard of scientific publication. Therefore, I would like to propose that the article should be revised thoroughly and need further reconsideration for re-evaluation whether this study article should be appropriate for acceptance or not.

1. What is the backgrond of this in vitro study. The ORIF of mandibular condyle fractures are well documented with some proved clinical evidence so far with various surgical approaches and various hardware materials. The back ground of this study must be well discribed using appropriate and recent references. The Introduction section and the Discussion sections are to be both the level of clinical reports of undergraduates, I'm sorry but, I need to say. The authors here also need to mention the ideal lines of double buttressing of mandibular condyle fracture should be well addressed in this section.

Response: Two paragraphs were re-written in ‘Introduction’ section as reviewer remarked. In ‘Discussion’ section, the ‘Ideal lines of double buttressing of mandibular condyle’ has been already described with some references. (Line 227-243)

- Line 32-45 : The subcondylar fracture commonly occurred at the sigmoid notch of mandible. Although, surgical approaches for open reduction risk facial nerve injury, it presented superior results to and fewer complications than closed reduction [1], for which present feasible visual field for accurate reduction with internal fixation devices such as mini-plates and screws [2]. However, the subcondyle, in particular, is a very difficult to be surgically approached that only present limited surgical access due to the presence of facial nerve. Therefore, surgeons’ experience and the fixation materials should be carefully considered for a successful clinical outcome. For the fixation materials, that requires mechanical strength as well as biocompatibility. Recently, with development of biomaterials for surgical treatment of mandible fractures, composites of unsintered hydroxyapatite particles and poly-l-lactide (u-HA/PLLA) has been used for open reduction, which reported reliable clinical outcome in mandibular body fracture [3,4].

However, Limited studies to date have reported on HA-PLLA fixation.

2. In the M&M section, all the materials tested for this research should be summarized including plate thickness, screw diameter, etc. "The thickness of the mini-plate was 2.0 mm, and the length and radius of the screw were 6.0 mm and 1.0 mm, respectively." is very unclear. The authors need to carry these photos as a figure. 1.0mm screw radius means 2.0mm diameter?? The surgeons don't describe the screw system with radius!

Response: As suggested, we have added Fig 1 and revised this sentence.

- Line 74-76 : The thickness of the mini-plate was 1.0 mm, and the length and diameter of the screw were 6.0 mm and 2.0 mm, respectively (Fig 1).

3. "The subcondylar fracture of FEA model was constructed that a 0.5 mm gap was set to minimize the effect of shear stress caused by the bony contact" seems hardly understandable. The authors need to further discuss this issue in the Discussion section well to persuade this idea. Actually the fractured segments are honestly well reduced as "buttressing as anatomical reduction" and the fixation should be done in a clinical setting. The present idea in in vito study should be against the clinical setting completely.

Response: Thank you for your comments. We made a new model without fracture gap, assuming bone reduction. Also, we re-analysis new model without fracture gap

4. Statistical analyses are mandatory. I've never reviewed the article without any statistical evaluation like this article, I'm afraid.

Response: Thank you for your suggestion. We have revised the Materials & methods section as below. In general, FEA constructs a single model and the data is extracted, so there is only one number of data. Therefore, statistical analysis was not performed in this study, and the data were compared based on the single data values.

- Line 119-125 : Our study used the finite element method to construct different models according to design factor and measure stress distribution and displacement for comparative analysis between various models. The finite element model for medical device analysis performs comparative analysis by changing specific designs or material properties except the same components such as cortical and cancellous bones [13-17]. Therefore, our study compared the single value for each material and did not use statistical analysis.

5. The discussion sections should be by far well discussed mentioning and comparing the results obtained here with those reported previously and with the clinical issues in a clinical setting. The comprehensive structures of the Discussion should be further amended. The clinical relevance should be well discribed and discussed based on the results obtained here.

Response: We appreciate reviewer’s detailed remarks over discussion. In observation of stress distribution on FEA model, biodegradable materials such as PLLA, HA-PLLA has less stress distribution, which meant they would inflict more stress than the metal materials on the biologic components of mandible. These results were consistent with other previous researches of Throckmorton and Meyer’s (reference 25,26). Therefore, we suggested the modification of plate and screw design according to the stress distribution in the materials (Line 252-276).

Response: We made conclusion from discussion with relevant references (ref. 2, 4, 6, 29, 30 and 31).

- Line 323-327 : For magnitude of deformation, a material with high stress distribution tended to experience less deformation, that reflected the results of stress distribution. The deformation in HA-PLLA at 600N less than 0.16 mm implicated that triangular fixation with two plates in subcondylar fracture could be expected favorable clinical outcome.

We made additional explanation for clinical relevance in line 238 as follows. 

Response: We made additional explanation for clinical relevance in Line 312 as follows.

- Line 313-316 : Subcondylar fracture is commonly unfavorable compared to mandibular body and angle fractures. Because, its dimension is relatively small and most of mastication stress would be concentrated on that. Therefore, the materials for subcondylar fracture should have appropriate mechanical strength as well as biocompatibility.

Could you please let me further re-review after substantial revisions are to be done for reconsideration for suitability?

#Reviewer2

Comments and suggestions

Summary of the research and my overall impression:

The aim of this research is particularly relevant to the field of Craniomaxillofacial Surgery because it is related to oral fractures, more specifically to mandible condylar ones. This subject is really challenging. Sometimes, surgeons don’t choose the open access to treat these fractures either due to anatomic reasons because important structures can be found there or due to the difficulty to reduce and to fix the fractures.

It is known that the open treatment is better than the closed one because it can facilitate the return to normal activities such as mastication and occlusion as soon as possible. Besides that, some complications caused by the non-surgical treatment can be avoided.

This paper shows a study about four kinds of ORIF to treat the mandible subcondylar fractures: titanium (Ti), magnesium alloy (Mg alloy), biodegradable polymers such as poly-L-lactic acid (PLLA) and a hydroxyapatite/poly-L-lactide (HA-PLLA). It is justified by the increasing demand of re-absorbable materials to be used instead of titanium, which is the gold standard material, owing to some disadvantages.

The methodology is especially interesting, based on other studies that have been published in high impact factor journals.

Abstract:

In order to be better understood by the readers of the journal, it would be interesting to follow the acronyms with the complete name of the materials. For example ti: titanium.

I suggest removing this sentence: “Deformation of HA-PLLA was approximately 0.15 mm upon 300 N of masticatory loading”. It is hard to understand in this section and it is well explained in the results section.

Response: Thank you for your suggestion. We have written Ti as titanium and removed the sentence “Deformation ~ loading.”

- Line 18-21 : Here, we simulated the mechanical properties of four fixation materials, namely, Titanium (Ti), Magnesium-alloy (Mg-alloy), poly-L-lactic acid (PLLA), and hydroxyapatite/poly-L-lactide (HA-PLLA), in a finite element analysis model of subcondylar fracture.

- Line 29 : We have removed this sentence

Introduction:

Line 10: “Subcondylar fractures can be treated conservatively or surgically”.

What does “conservatively” mean? The best term may be “non-surgically”, as we can have a conservative surgery, for example, depending on our care during the procedure.

Response: We revised the introduction section.

- Line 32-44 : The subcondylar fracture commonly occurred at the sigmoid notch of mandible. Although, surgical approaches for open reduction risk facial nerve injury, it presented superior results to and fewer complications than closed reduction [1], for which present feasible visual field for accurate reduction with internal fixation devices such as mini-plates and screws [2]. However, the subcondyle, in particular, is a very difficult to be surgically approached that only present limited surgical access due to the presence of facial nerve. Therefore, surgeons’ experience and the fixation materials should be carefully considered for a successful clinical outcome. For the fixation materials, that requires mechanical strength as well as biocompatibility. Recently, with development of biomaterials for surgical treatment of mandible fractures, composites of unsintered hydroxyapatite particles and poly-l-lactide (u-HA/PLLA) has been used for open reduction, which reported reliable clinical outcome in mandibular body fracture [3].

Materials and methods:

In Figure 1, the posterior plate does not seem to be completely attached to the bone. I would like you to confirm this and try to explain it better.

The second screws (considering the superior to inferior) are closer to the fracture line than the ideal. On the anterior plate, this screw seems to be in the fracture line. Depending on the direction of the perforation to install this screw, I think that the fracture can be “open”. Maybe, this fact can be considered a bias.

I don’t think it can influence the results, but I think that you can try to reinstall this plate, virtually.

I suggest you provide more information about the masticatory load that was applied. Why did you choose this load? Isn´t it too high? I would like to see other papers that justify this choice.

Response: Thank you for your comments. As you suggested, we have repositioned the plate position and re-performed the analysis. Please find Figure 2 and Figure 5 for the plate position. 

Also, We set up occlusal forces 132N at one week and 300N at 6 weeks after surgery. One week and 6 weeks after surgery are important timing in mandible surgery, for patients would commonly start to have food intake under elastic occlusal guidance. In healthy adult, usual mastication force is 700N (that was explained Line 234-243). Clinically relevant mastication force during formation of primary bony callus in patients who had mandible surgery usually would be observed from one week to 6 weeks after surgery, though we simulated deformation of materials to extreme condition such as 1000N. 

These set up based on other previous studies as below. (we’ll added these two references in revised manuscript). 

J Oral Maxillofac Surg

. 2000 Apr;58(4):370-3; discussion 373-4. doi: 10.1016/s0278-2391(00)90913-3.

Measure of bite force and occlusal contact area before and after bilateral sagittal split ramus osteotomy of the mandible using a new pressure-sensitive device: a preliminary report

K Harada 1, M Watanabe, K Ohkura, S Enomoto

J Oral Maxillofac Surg

. 2014 Feb;72(2):402.e1-13. doi: 10.1016/j.joms.2013.10.003. Epub 2013 Oct 17.

Biomechanical evaluation of magnesium-based resorbable metallic screw system in a bilateral sagittal split ramus osteotomy model using three-dimensional finite element analysis

Jin-Yong Lee 1, Jung-Woo Lee 2, Kang-Mi Pang 3, Hyoun-Ee Kim 4, Soung-Min Kim 5, Jong-Ho Lee 6

- Line 234-236 : We assumed 132 N of force at 1 week after surgery, 300 N at 6 weeks after surgery, and 700 N in the healthy adult’s mastication loading based on previous studies [4,27,28].

Results:

In Figure 2: “middle column, stress on fixations (mini-plates and screws) but we can just see the plates, without screws”.

I think that you need to change this caption due to the fact that the screws are not there, just the plates.

Do you have information about the bone? If so, you can insert main maximum tension and minimum tension to increase the amount of information and to make this article even more relevant.

Response: As suggested, we have added maximal and minimal tensile stress values and revised the middle column in Fig 3.

- Line 144-152 : The tensile stress distribution of cortical bone was highest for Mg-alloy, followed by Ti, HA-PLLA and PLLA. The stress distribution was similar between Ti and Mg-alloy and between HA-PLLA and PLLA (Table 2 and Table 3). On the other hand, the tensile stress distribution of cancellous bone was highest for Mg-alloy, followed by HA-PLLA, Ti and PLLA. The stress distribution of screw hole in the bones was higher in the anterior screw hole compared to the posterior hole. Facing the fracture line, the contact area of the screw hole was applied compressive stress, and tensile stress occurred on the opposite side (Fig 3a,d,g,j).

---

## [Decision Letter · Decision Letter 1]

27 Aug 2020

PONE-D-20-14876R1

Biomechanical evaluation of unilateral subcondylar fracture of the mandible on the varying materials: a finite element analysis

PLOS ONE

Dear Dr. Kim,

Thank you for submitting your manuscript to PLOS ONE. After careful consideration, we feel that it has merit but does not fully meet PLOS ONE’s publication criteria as it currently stands. Therefore, we invite you to submit a revised version of the manuscript that addresses the points raised during the review process.

ACADEMIC EDITOR: Authors should improve English language by native English speaker for the whole text of the paper before consideration of publication />==============================

We look forward to receiving your revised manuscript.

Kind regards,

Essam Al-Moraissi

Academic Editor

PLOS ONE

Reviewers' comments:

Reviewer's Responses to Questions

**Comments to the Author**

1. If the authors have adequately addressed your comments raised in a previous round of review and you feel that this manuscript is now acceptable for publication, you may indicate that here to bypass the “Comments to the Author” section, enter your conflict of interest statement in the “Confidential to Editor” section, and submit your "Accept" recommendation.

Reviewer #1: (No Response)

Reviewer #2: All comments have been addressed

2. Is the manuscript technically sound, and do the data support the conclusions?

Reviewer #1: No

Reviewer #2: Yes

3. Has the statistical analysis been performed appropriately and rigorously? 

Reviewer #1: No

Reviewer #2: N/A

4. Have the authors made all data underlying the findings in their manuscript fully available?

Reviewer #1: No

Reviewer #2: Yes

5. Is the manuscript presented in an intelligible fashion and written in standard English?

Reviewer #1: No

Reviewer #2: Yes

6. Review Comments to the Author

Reviewer #1: I here carefully re-evaluated the revised set of the manuscript and found it no merits of publication. I'm sorry but I can't support this study article. Further the re-submitted article would be wrongly carried with English grammatical corrections.

Reviewer #2: Thanks for answering my questions. I think that the information now is better and you improved your paper.I appreciate it.

7. PLOS authors have the option to publish the peer review history of their article (what does this mean?). If published, this will include your full peer review and any attached files.

Reviewer #1: No

Reviewer #2: **Yes: **Ricardo Augusto Conci

---

## [Author Response · Author response to Decision Letter 1]

10 Sep 2020

#Reviewer1

Comments and suggestions

Reviewer #1: I here carefully re-evaluated the revised set of the manuscript and found it no merits of publication. I'm sorry but I can't support this study article. Further the re-submitted article would be wrongly carried with English grammatical corrections.

Response: Thank you for your suggestion.

We have undergone the service by Editage (www.editage.co.kr) for English language editing. I attached the certification.

Moreover, our study examined simulations of four different fixation materials (titanium, magnesium alloy, poly-L-lactic acid [PLLA], and hydroxyapatite/poly-L-lactide [HA-PLLA]) in finite-element analysis models of subcondylar mandibular fracture. 

Predicted result using computer simulation (mathmetical annalysis) was affected only by change of material’s property (Elastci Modulus, Poisson’s Ratio) under same conditions (design, fracture type, load condition etc.). therefore, repeated simulation result is always same for same material with same condition.

Reviewer #2: Thanks for answering my questions. I think that the information now is better and you improved your paper. I appreciate it.

Response: Thank you for your kind comment.

---

## [Decision Letter · Decision Letter 2]

25 Sep 2020

Biomechanical evaluation of unilateral subcondylar fracture of the mandible on the varying materials: a finite element analysis

PONE-D-20-14876R2

Dear Dr. Kim,

We’re pleased to inform you that your manuscript has been judged scientifically suitable for publication and will be formally accepted for publication once it meets all outstanding technical requirements.

Kind regards,

Essam Al-Moraissi

Academic Editor

PLOS ONE

Additional Editor Comments (optional):

Reviewers' comments:

Reviewer's Responses to Questions

**Comments to the Author**

1. If the authors have adequately addressed your comments raised in a previous round of review and you feel that this manuscript is now acceptable for publication, you may indicate that here to bypass the “Comments to the Author” section, enter your conflict of interest statement in the “Confidential to Editor” section, and submit your "Accept" recommendation.

Reviewer #2: All comments have been addressed

Reviewer #3: All comments have been addressed

2. Is the manuscript technically sound, and do the data support the conclusions?

Reviewer #2: Yes

Reviewer #3: Yes

3. Has the statistical analysis been performed appropriately and rigorously? 

Reviewer #2: N/A

Reviewer #3: (No Response)

4. Have the authors made all data underlying the findings in their manuscript fully available?

Reviewer #2: No

Reviewer #3: Yes

5. Is the manuscript presented in an intelligible fashion and written in standard English?

Reviewer #2: Yes

Reviewer #3: Yes

6. Review Comments to the Author

Reviewer #2: (No Response)

Reviewer #3: The design of this study is very unique and interesting.

The data is clear and the manuscript is well corrected.

It seems that there is no problem as a paper to be published.

7. PLOS authors have the option to publish the peer review history of their article (what does this mean?). If published, this will include your full peer review and any attached files.

Reviewer #2: **Yes: **Ricardo Augusto Conci

Reviewer #3: No

---

## [Editor Report · Acceptance letter]

29 Sep 2020

PONE-D-20-14876R2 

Biomechanical evaluation of unilateral subcondylar fracture of the mandible on the varying materials: a finite element analysis 

Dear Dr. Kim:

I'm pleased to inform you that your manuscript has been deemed suitable for publication in PLOS ONE. Congratulations! Your manuscript is now with our production department. 

Kind regards, 

on behalf of

Dr. Essam Al-Moraissi 

Academic Editor

PLOS ONE